# The Role of Astrocytes in the Mechanism of Perioperative Neurocognitive Disorders

**DOI:** 10.3390/brainsci12111435

**Published:** 2022-10-25

**Authors:** Ying Cao, Xiaowan Lin, Xiao Liu, Kang Yu, Huihui Miao, Tianzuo Li

**Affiliations:** Department of Anesthesiology, Beijing Shijitan Hospital, Capital Medical University, 10th Tieyi Road, Haidian District, Beijing 100038, China

**Keywords:** astrocytes, perioperative neurocognitive disorder, mechanism

## Abstract

Recently, astrocytes are fast climbing the ladder of importance in cognitive-related diseases. Perioperative neurocognitive disorder (PND) is a common consequence of anesthesia and surgery, which is widely investigated in elderly and susceptible individuals. There is no doubt that astrocytes also play an irreplaceable role in the pathogenesis of PND. Reactive astrocytes can be found in the PND model, with an altered phenotype and morphology, suggesting a role in the development of the diseases. As a prominent participant cell in the central inflammatory response, the inflammatory response is unavoidably a crucial pathway in the development of the disease. Astrocytes also play a significant role in the homeostasis of the internal environment, neuronal metabolism, and synaptic homeostasis, all of which have an impact on cognitive function. In this article, we discuss the function of astrocytes in PND in order to establish a framework for investigating treatments for PND that target astrocytes.

## 1. Introduction

Disordered neurocognitive functions after surgery and anesthesia is a heterogeneous set of situations, which include any form of the acute event (of postoperative delirium) and cognitive decline diagnosed up to 30 days after the procedure (a delayed neurocognitive recovery) and up to 12 months (as a postoperative neurocognitive disorder, POCD) [1,2] (p. 11). Previously, all forms of the impairment were called POCD, but in November 2018, six journals published the same article suggesting the naming of all types of cognitive impairment associated with the perioperative period as perioperative neurocognitive disorders (PND) [1] (p. 11). PND represents an acute or persistent cognitive impairment, including memory, attention, information processing, and cognitive flexibility [3] (p. 11). Age has been identified as a major risk factor for PND in previous studies [4] (p. 11). The incidence of PND in middle-aged patients is 19.2% [5] (p. 11) and up to 52% in senior patients [6,7] (p. 11). The global population of old people has grown considerably; as a result, it is reasonable to expect that the number of PND cases will continue to rise. PND causes longer hospital stays, considerably higher mortality rates within 6-12 months following surgery, and greater medical expenditures (acting as a heavy burden on families and society).

There are several hypotheses on the pathogenesis of PND. The first, and the most extensively explored, is neuroinflammation [8] (p. 11). Multiple avenues allow for peripheral inflammatory substances after surgery to enter the brain. It ultimately results in a cerebral inflammatory response [9,10] (pp. 11–12). Thus, in the PND model, pro-inflammatory signaling molecules can be found in both the CNS and peripheral nervous system [11] (p. 12). Secondly, protein abnormalities, including Aβ and Tau proteins, have also been linked to the development of PND [12,13] (p. 12). We all know that cognitive impairment can result from damage to synaptic plasticity, which is required for learning and memory [14,15] (p. 12). In addition, neurotransmitter imbalance is crucial in PND. Despite accumulating research on PND, the specific processes through which cognitive deterioration develops following surgery and anesthesia remain uncertain.

Recent research has found that astrocytes play a variety of roles in the CNS. Astrocytes participate in immunological responses, regulate neurotransmitters and calcium homeostasis, modulate synapse formation, maturation, and elimination, maintain the blood–brain barrier, control ion homeostasis, and brain blood flow, and provide nutritional and trophic support to the brain [16] (p. 12). Notably, astrocytes are involved in many cognitive illnesses, such as depression [17] (p. 12) and schizophrenia [18] (p. 12), as well as the majority of neurodegenerative diseases, including Alzheimer’s disease (AD) [19] (p. 12), Parkinson’s disease (PD) [20] (p. 12), and Huntington’s disease (HD) [21] (p. 12). As an important component dysregulated in cognitive impairment, the role and molecular mechanism of dysfunctional astrocytes in the pathogenesis of PND are explored.

## 2. Different Phenotypes of Reactive Astrocytes and PND

It is of growing interest that astrocytes subject to different injuries develop into distinct subtypes. Astrocytes are classified into A1 and A2 states in the majority of studies; A1 exhibits a detrimental and proinflammatory phenotype, while A2 promotes repair [22,23] (p. 12). Active microglia stimulated by lipopolysaccharides (LPS) could produce a variety of pro-inflammatory cytokines, such as interleukin-1 (IL-1), tumor necrosis factor α (TNF-α), and complement C1q (C1q), which would trigger the polarization of A1 astrocytes [24] (p. 12). An unknown process of development induces A2 astrocytes in an ischemia scenario [25] (p. 13). A1 astrocytes are extremely neurotoxic and lose several characteristics of the normal astrocyte, including synapse formation and function, as well as synaptocytosis [24] (p. 12). A2, on the other hand, upregulates neurotrophic or anti-inflammatory genes that enhance neuronal survival, implying a protective effect on the nervous system [25] (p. 13). One article has already reported that component C3 (a major A1 astrocyte-specific marker protein) is upregulated in the CNS in a mouse model of orthopedic surgery-induced PND. Since C3 expression is elevated mainly in hippocampal astrocytes, it is hypothesized that A1-type astrocytes might be involved in the pathogenesis of PND [26] (p. 13). Another article clearly states that microglia activation in the hippocampus generates A1 neurotoxic astrocytes, which then facilitate etomidate to induce PND [27] (p. 13). In addition, suppressing A1 astrocytes with octreotide cholecystokinin (CCK-8) enhances hippocampus glutamatergic synapse development and postoperative cognitive functions in aged mice [28] (p. 13). Most unexpectedly, the latest article reports that esketamine administration could alleviate PND post-surgery in aged rats by down-regulating neuronal Aβ-42 and up-regulating hippocampal type A2 astrocytes [29] (p. 13). Present data suggest that researchers have identified that changes in astrocyte phenotype will have a significant impact on PND. However, it is unclear how A1 and A2 phenotypes and reactive astrocytes interact with each other in the PND of aged mice.

Aside from function, astrocyte morphology also changes upon activation [30] (p. 13). Alterations in astrocyte morphology have been observed in the general anesthesia model [31] (p. 13). Additionally, other examples from the literature have reported that the sevoflurane-inhibited Ca^2+^ channel in astrocytes results in the downregulation of Ezrin, a membrane protein bound to actin. Ezrin is linked to astrocyte morphogenesis. Inducing Ezrin overexpression within astrocytes rescues astrocytic and neuronal dysfunctions and fully corrected social behavior deficits in sevoflurane group mice [32] (p. 13). Changes in the astrocyte morphology are valuable resources for a thorough analysis of the effects of astrocytes on PND.

## 3. Astrocytes Are Involved in the Development of PND Partially by Mediating Neuroinflammation

Presently, many clinical and preclinical studies have confirmed the correlation between neuroinflammation and PND [33] (p. 13). Surgical trauma and pain induce systemic inflammatory responses and the release of systemic inflammatory mediators, which enter the central nervous system via the relatively permeable blood–brain barrier and activate the microglia and astrocyte to secrete additional cytokines, creating a central inflammatory state [34] (p. 13). More than one article has reported that different stages of pain exacerbate an inflammatory response that affects PND, including acute and chronic pain in the preoperative period and acute pain in the postoperative period [35,36] (p. 13). Astrocytes act as important regulators for neuroinflammation. As shown in Figure 1, astrocytes can promote inflammation through the recruitment of peripheral inflammatory cells, the activation of CNS-resident microglia, and their intrinsic neurotoxic activities. Eventually, they can lead to an uncontrollable inflammatory response [37,38] (p. 13).

Recent evidence demonstrates that the C–C motif ligand 2 (CCL2) plays a vital role in the nervous system and is predominantly derived from astrocytes, while CCR2 (the main receptor for CCL2) is predominantly expressed in microglia. The surgery increases chemokine CCL2 release from the activated astrocytes, which promotes microglia activation and M1 polarization. M1 microglia is one of the phenotypes formed by stimulated microglia, which is considered to be destructive and can produce inflammatory mediators. Cellular interactions promote inflammatory responses and cause hippocampus neuronal injuries, jointly resulting in the development and progression of cognitive impairment [39] (p. 13). In addition to microglia, mast cells can also interact with astrocytes and contribute to PND [40,41] (p. 13).

The phenotype of astrocytes is a concept that has emerged in the last few years. Several investigators investigating the astrocyte induction of PND via an inflammatory response did not distinguish which phenotype; instead, most authors merely reported that the astrocytes were in an activated state following anesthesia and surgery.

An earlier paper reported that PND is associated with IL17A-induced astrocyte activation and that there is a consequent TGFβ/Smad pathway-dependent secretion of Aβ_1–42_ in the hippocampus [42] (p. 13). A recent study found that after surgery, there is an activation of astrocytes and lipid accumulation in the hippocampus. However, the activation of cannabinoid receptor type II (CB2R) significantly reduces astrocyte lipid accumulation and decreases the release of the inflammatory factors of interleukin-1 beta (IL-1β) and interleukin 6 (IL-6) in the peripheral serum while improving cognitive performance in the PND mice [43] (p. 13). Recently, increasing evidence has proven that the locus coeruleus noradrenergic (LCNE) system participates in regulating neuroinflammation in some neurodegenerative disorders. In a study, N-(2-chloroethyl)-N-ethyl-2-bromobenzylamine (DSP-4) was injected intracerebroventricularly into each rat before anesthesia/surgery to deplete the locus coeruleus (LC) noradrenaline (NE). Pretreatment with DSP-4 decreased the levels of systemic and central NE, decreased the level of interleukin-1β (IL-1β) in the hippocampus, and attenuated hippocampal-dependent learning and memory impairment in the rats with PND, with a downregulation in the activation of astrocytes [44] (p. 14). A number of papers have reported that limiting astrocyte activation or reducing the inflammatory response is beneficial to PND. For example, electroacupuncture is found to improve cognitive impairment and inhibit astrocyte activation and oxidative stress in the hippocampus of aged rats with PND [45] (p. 14). Edaravone could not only inhibit the over-activation of astrocytes but also decrease the expression of pro-inflammatory cytokines, indicating that edaravone can alleviate surgery-induced neuroinflammation in aged animals [46] (p. 14).

## 4. Abnormal Astrocytes Induce PND by Regulating Synapse Formation, Quantity, and Transmission of Neurotransmitters

Synapses are specific structures that contact neurons and glial cells in the CNS. Synapses are involved in the transportation of ions and neurotransmitters as well as the exchange of neuronal information [47] (p. 14). As shown in Figure 2, it is quite clear that astrocytes, along with the pre-and post-synaptic nerve terminals, construct a triad of synapses that enable the formation and conduction of neuronal signals [48] (p. 14). Firstly, astrocytes release components such as thrombospondin (TSP), Hevin, and secreted proteins acidic and rich in cysteine (SPARC), which participate in synapse development and synaptic activity regulation. β-amyloid (Aβ) is an important cause of PND, while TSP can inhibit the degenerative alterations induced by Aβ [49,50] (p. 14). The likelihood that astrocytes trigger PND via TSP is quite strong. Secondly, in both in vivo and in vitro models, astrocytes play an important role in synaptic phagocytosis throughout the brain [51,52] (p. 14). Astrocytes can wrap around dystrophic neurons and devour them, which is one benefit of synaptic phagocytosis, but excessive phagocytosis might accelerate synaptic loss. PND has been shown to be associated with the dysregulation of phagocytosis, although this study primarily describes microglia involvement and ignores astrocytes [53] (p. 14). There is no doubt that the brain’s phagocytosis is primarily controlled by microglia. However, astrocytes can influence microglia-mediated synaptic pruning by inducing the activation of the complement cascade. The complement system, which is a component of the immune system, can control inflammatory reactions and take part in synaptic pruning [54] (p. 14). The C3 plays a central role in complement system activation. An increased release of C3 from the hippocampal astrocytes in PND has been shown in the experiments of our group and the other literature, and an increase in C3 may lead to excessive synaptic pruning and, therefore, cognitive impairment [26] (p. 13).

Finally, astrocytes control neurotransmitter transmissions, see Figure 2. Glutamate, the primary excitatory neurotransmitter in the mammalian brain, is necessary for cognitive functions. A low level of extracellular glutamate is maintained by astrocytes, which remove roughly 90% of glutamate via glutamate uptake transporters [55] (p. 14). EAAT1/GLAST and EAAT2/GLT1 are the major glutamate transporters. It has been found that an up-regulation of GLAST to try to maintain low extracellular glutamate levels in aged rats is associated with isoflurane-induced memory impairment, but hippocampal glutamates are still raised [56] (p. 14). Glutamine synthetase (GS) [57] (p. 14) and glutamate dehydrogenase 1 (GLUD1) [58] (p. 14), which are two astrocytic enzymes modulating the level of extracellular glutamate, are changed in aged rats with PND. Glutamine synthetase (GS) is decreased and causes glutamate toxicity in aged rats with PND [59] (p. 14).

In contrast, recent studies have revealed that in addition to uptake, astrocytes release trace amounts of glutamate to the adjacent neurons. This release of astrocytic glutamate is plausibly mediated by Ca^2+^-dependent exocytosis, but other mechanisms have also been proposed to mediate the release of glutamate from astrocytes as well [60] (p. 14). It has already been reported that the excessive release of inflammatory transmitters, such as tumor necrosis factor-α (TNF-α) and prostaglandin E (PGE), contribute to an increase in Ca^2+^ in astrocytes and neuronal excitotoxicity induced by an excessive glutamate release [61] (p. 15). These results imply that the levels of extracellular glutamate control by astrocytes play a significant role in the pathophysiology of PND.

Acetylcholine is also known to be one of the major neurotransmitters of the central nervous system. Astrocytes in the hippocampus express both nicotinic and muscarinic cholinergic receptors [62,63] (p. 15) and respond to the local release of acetylcholine with increased Ca^2+^ concentrations [63] (p. 15). These processes may contribute to the cholinergic regulation of neurosynaptic transmission and memory. A recent study confirmed that surgery impairs the muscarinic receptor activation-evoked Ca^2+^ influx in hippocampal astrocytes, which leads to cognitive impairment [64] (p. 15).

In addition, D-serine, which interacts with the NMDA receptor’s glycine site and co-activates the NMDA receptor with the agonists [65] (p. 15), is involved in a variety of physiological and pathological processes, including neurodevelopment, synaptic plasticity, learning, memory, pain, and neurodegenerative disorders [66] (p. 15). Under normal settings, astrocytes produce relatively little serine racemase (SR) to catalyze the generation of D-serine and make minimal contributions to synaptic plasticity. However, activated and reactive astrocytes overexpress SR [67] (p. 15) to generate D-serine, which then causes neurotoxicity. Importantly, traumatic brain injuries cause the reactive activation of astrocytes in the hippocampus as well as a transition in D-serine syntheses from neurons to astrocytes. Damage to synaptic plasticity or learning might be reversed when SR is eliminated from the reactive astrocytes [68] (p. 15). Although there is no clear indication as to the association between D-serine and PND, there is reason to assume the use of D-serine as a target for PND in terms of neurotoxicity and synaptic plasticity.

## 5. Astrocytes Maintain Homeostasis of Intracerebral Microenvironment

Firstly, astrocytes are a major component of the blood–brain barrier (BBB). Astrocytes can send processes to the vasculature, and these processes terminate with endfeet, which covers up to 98% of the entire vascular surface [69,70] (p. 15), as shown in Figure 1. Endfeet are filled with aquaporin 4 (AQP4) water channels [71] (p. 15). On the one hand, surgery could disrupt the expression of aquaporin hence damaging the BBB [72] (p. 15) and allowing the entry of peripheral immune cells to the CNS. On the other hand, the AQP is a component of the brain’s waste clearance system and is essential for the functioning of the glymphatic system. When AQP4 is destroyed, it can lead to the accumulation of protein waste in the brain, such as Aβ, which activates microglia and triggers neuroinflammation [73] (p. 15). The upregulation of AQP-4 in astrocytes improves the clearance of Aβ deposits following anesthesia and surgery in aged mice [74] (p. 15). Several studies have linked waste or protein aggregation caused by glymphatic dysfunction to neurodegenerative diseases like Alzheimer’s disease [75] (p. 15). Anesthesia and surgery could induce the disruption of the glymphatic system constructed by astrocytes, resulting in a build-up of waste products that could trigger or worsen neuroinflammation, eventually leading to cognitive impairment. Ren et al. summarized the evidence regarding the lymphatic dysfunction of the PND model, offering novel treatment options for the disease [73] (p. 15).

Secondly, in physiological conditions, astrocytes are coupled with gap junctions, which facilitate intercellular signaling in the parenchymal portion of the BBB. Gap junctions are composed mainly of the protein connexin 43 (Cx43). Since chronic isoflurane anesthesia reduces the expression of GJs-Cx43 in the hippocampus, the reduction in GJs-Cx43 levels may further lead to defects in astrocyte networks initiating or exacerbating neuroinflammation and ultimately leading to cognitive impairment in mice [76] (p. 15).

Thirdly, in pathological conditions associated with the BBB breakdown, the endfeet of reactive astrocytes can express tight junction proteins to mount a parenchymal line of defense [77] (p. 15). A number of articles have been published which demonstrate the reduction in tight junction proteins in the PND model [78,79] (p. 15). However, there is no clear indication of whether the tight junction proteins associated with PND are synthesized by astrocytes. A recent paper, however, suggested that the tight junction proteins claudin 1, claudin 4, and junctional adhesion molecule-A were synthesized by astrocytes and played an important role in inflammatory CNS injuries [80] (p. 16). Therefore, it is reasonable to assume that the tight junction proteins synthesized by astrocytes will also have some effect on PND.

Lastly, astrocytes secrete numerous paracrine factors affecting the barrier properties of the BBB, such as morphogens Sonic hedgehog (Shh), vascular endothelial growth factor (VEGF), apolipoprotein E (ApoE) and others [81,82] (p. 16).Shh, when secreted by reactive astrocytes, promote the integrity of the BBB after injury by interacting with the Hedgehog (Hh) receptors in endothelial cells [83] (p. 16). It has been shown that isoflurane induces the activation of VEGF proteins and the disruption of the blood–brain barrier in the hippocampus. The inhibition of the isoflurane-induced VEGF protein by hyperactivation may be relevant to rescue learning memory deficits in aged rats [84] (p. 16).

Astrocytes represent a major source of apolipoprotein E (ApoE) in the brain. The expression of the human APOE4 isoform is a risk factor for PND [85] (p. 16). There are experiments that clearly show the rescue of the BBB phenotype at both a structural and functional level by reducing astrocyte-produced ApoE4 [86] (p. 16). However, there are also studies that show that a complete knockdown of ApoE from birth leads to BBB leakage [87] (p. 16). It is valuable to think about what effect this would have on a PND model when reducing astrocyte-produced ApoE4.

## 6. Astrocytes Assist Neurons in Trophic Metabolism

The human brain accounts for only 2% of the overall body weight, but it consumes approximately 20% of energy at rest. Energy expenditure in the brain is boosted even more by various tasks [88] (p. 16). This relatively efficient energy processing in the brain relies on the metabolic plasticity of astrocytes. Astrocytes are anatomically located between dense neuronal structures and complex branches of the brain’s vasculature and can influence energy metabolism in the brain in a variety of ways [89] (p. 16).

Firstly, astrocytes regulate cerebral blood flow [90] (p. 16). Disruption to the normal physiology of astrocytes can compromise blood flow regulation, resulting in pathological circumstances like Alzheimer’s disease [91] (p. 16). It is probable that low cerebral blood flow perfusion is a major cause of PND [92,93] (p. 16), while increasing the blood supply to the brain could be beneficial.

In addition to regulating blood flow, astrocytes can transport lactates to neurons, see Figure 2. Lactate is transported between the neurons and astrocytes to provide energy for brain activity and metabolism. After peripheral surgery, the metabolic environment of the hippocampus is changed, while the absence of glucose and lactate transporters in the astrocytes is accompanied by alterations to lactate concentrations, which culminate in neurodegenerative diseases, such as PND [94] (p. 16). A recent study reported that lactate decreases between 6 and 72 h after peripheral surgery, which leads to neuronal dysfunction [95] (p. 16). Different anesthetic agents inhibit astrocytic glycolysis and reduce the level of extracellular lactates in the brain [96] (p. 16). Other research suggests that interfering with the lactate transport from astrocytes to neurons inhibits synaptic plasticity and memory, which supports the functional role of lactates for postoperative cognition [97,98] (p. 16).

In addition, astrocytes secrete glial cell line-derived neurotrophic factors (GDNF), which are fundamental in neurogenesis and learning. Amantadine reduces cognitive impairment following anesthesia and surgery by increasing the GDNF level [99] (p. 17). In a clinical trial, serum GDNF levels were lower in patients with PND at 2 and 7 days postoperatively than in the healthy controls, implying that GDNF might participate in the development of PND [100] (p. 17). Accordingly, GDNF overexpression improves cognitive function in the cognitive deficit model of aged rats and neonatal rats [101,102] (p. 17).

## 7. Conclusions

In summation, astrocytes can influence cognitive performance following anesthesia and surgery via a variety of mechanisms (Table 1), while some of the mechanisms are unidentified. As a result, more research on astrocytes might lead to new ideas for determining the pathophysiology of PND, as well as new directions for PND therapy.

## Figures and Tables

**Figure 1 brainsci-12-01435-f001:**
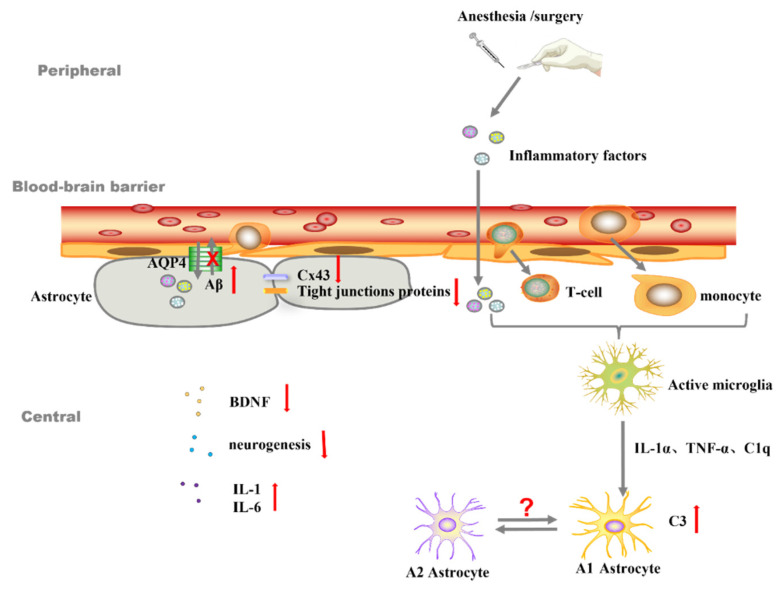
The role of astrocytes in neuroinflammation and blood–brain barrier. IL-1α: interleukin-1α. TNF-α: tumor necrosis factor α. C1q: component 1q. C3: Component C3. Aβ: β-amyloid. Cx43: Gap junctions-connexin 43. BDNF: Brain-derived neurotrophic factor.

**Figure 2 brainsci-12-01435-f002:**
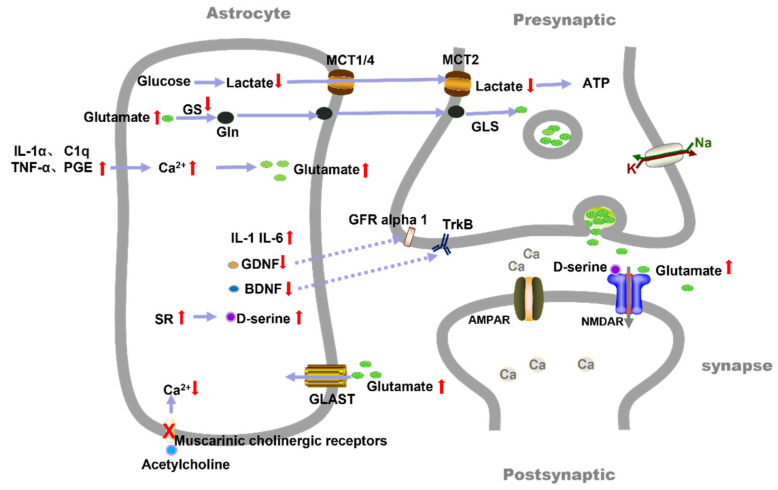
The role of astrocytes in synapse and trophic metabolism. GLUT1/3: glucose transporter 1/3. GLU: Glutamate. GS: Glutamine synthetase. Gln: glutamine. GLS: Glutaminases. GDNF: glial cell line-derived neurotrophic factor. GLT-1/GLAST: glutamate transporters. MCT1/2/4: monocarboxylate transporters 1/2/4. SR: serine racemase. TNF-α: Tumor necrosis factor-α. PGE: prostaglandin E. BDNF: Brain-derived neurotrophic factor. TrkB: BDNF receptors. GFR alpha 1: GDNF receptors.

**Table 1 brainsci-12-01435-t001:** The implication of astrocytes in PND. C3: Component C3. CCL2: chemokine C-C motif ligand 2. TSP: thrombospondin. GLAST: glutamate transporters. GS: Glutamine synthetase. GLUD1: glutamate dehydrogenase 1. SR: serine racemase. BBB: blood–brain barrier. AQP-4: aquaporin4. GDNF: glial cell line-derived neurotrophic factor. Shh: Sonic hedgehog. VEGF: vascular endothelial growth factor. ApoE: apolipoprotein E.

The Implication of Astrocytes in PND		References
phenotypes of reactive astrocytes	A1 astrocyte is upregulated in the CNS. Suppressing A1 astrocytes enhances postoperative cognitive function in aged mice. Up-regulating hippocampal type A2 astrocytes could alleviate PND.	[27,28,29]
	Alterations in astrocyte morphology have been observed in the general anesthesia model. Sevoflurane causes downregulation of Ezrin, which is associated with astrocyte morphogenesis. Induction of Ezrin overexpression in astrocytes corrects social behavior deficits in sevoflurane-group mice.	[31,32]
Mediating neuroinflammation	Surgery increases CCL2 release from activated astrocytes, which promotes microglia activation and M1 polarization.	[32]
	Mast cells interact with astrocytes and contribute to PND.	[37,38]
	PND is associated with IL17A-induced astrocyte activation.	[39]
	Surgery leads to activation of astrocytes and lipid accumulation. Activation of cannabinoid receptor type II (CB2R) decreases the release of the inflammatory factors.	[40]
	Depletion of the locus coeruleus (LC) noradrenaline (NE) decreases the level of interleukin-1β (IL-1β) and attenuates learning and memory impairment in rats with PND, with a downregulation in the activation of astrocytes.	[41]
	Electroacupuncture can improve cognitive impairment and inhibit astrocyte activation and oxidative stress in the hippocampus of aged rats with PND.	[42]
	Edaravone could not only inhibit the over-activation of astrocytes, but also decrease the expression of pro-inflammatory cytokines.	[43]
Regulating synapse formation, quantity, and transmission of neurotransmitters	β-amyloid (Aβ) is an important cause of PND, and TSP released from astrocyte can inhibit Aβ-induced degenerative changes.	[46,47]
	PND has been shown to be associated with dysregulation of phagocytosis. Astrocytes can influence microglia-mediated synaptic pruning by inducing the activation of the complement cascade. An increased release of C3 from hippocampal astrocytes in PND may lead to excessive synaptic pruning and therefore cognitive impairment.	[26,49,50]
	Glutamate is the primary excitatory neurotransmitter in the mammalian brain. A low level of extracellular glutamate is maintained by astrocytes. It has been found that an up-regulation of GLAST to try to maintain low extracellular glutamate levels in aged rats is associated with isoflurane-induced memory impairment, but hippocampal glutamates are still raised. GS and GLUD1 are changed in aged rats with PND. GS is decreased and causes glutamate toxicity in aged rats with PND.	[52,53,54,55,56]
	Acetylcholine is also known to be one of the major neurotransmitters of the central nervous system. Astrocytes respond to local release of acetylcholine with increased Ca^2+^ concentrations Surgery impairs muscarinic receptor activation-evoked Ca^2+^ influx in hippocampal astrocytes, which leads to cognitive impairment.	[60,61]
	Activated astrocytes overexpress SR and generate D-serine which then cause neurotoxicity. Damage to synaptic plasticity or learning might be reversed when SR is eliminated from the reactive astrocytes.	[64,65]
Maintain homeostasis of intracerebral microenvironment	Surgery could disrupt the expression of aquaporin hence the BBB is damaged, allowing entry of peripheral immune cells to the CNS. Impaired AQP can lead to the accumulation of protein wastes in the brain, such as Aβ. AQP is essential for the functioning of the glymphatic system. Anesthesia and surgery could induce disruption of glymphatic system constructed by astrocytes, resulting in a build-up of waste products that could trigger or worsen neuroinflammation, eventually leading to cognitive impairment.	[69,70]
	Chronic isoflurane anesthesia reduces expression of GJs–Cx43 in the hippocampus, and ultimately leads to cognitive impairment in mice.	[73]
	Articles have demonstrated the reduction in tight junction proteins in the PND model. Although there is no clear indication whether the tight junction proteins associated with PND are synthesized by astrocytes., a recent paper suggested that tight junction proteins synthesized by astrocytes play an important role in inflammatory CNS injury.	[75,76,77]
	Astrocytes secrete numerous paracrine factors affecting barrier properties of the BBB, such as morphogens Sonic hedgehog (Shh), vascular endothelial growth factor (VEGF), apolipoprotein E (ApoE), and others. Shh secreted by reactive astrocytes promotes the integrity of the BBB after injury by interacting with Hedgehog (Hh) receptors in endothelial cells. Isoflurane induces activation of VEGF proteins and disruption of the BBB. The inhibition of isoflurane-induced VEGF protein hyperactivation may be relevant to rescue learning memory deficits in aged rats. Rescue of the BBB can be achieved by reducing astrocyte-produced ApoE4. However, complete knockdown of ApoE from birth leads to BBB leakage.	[78,79,80,81,82,83]
Assist neurons in trophic metabolism	It is probable low cerebral blood flow perfusion is a major cause of PND. Disruption of normal physiology in astrocytes can compromise blood flow regulation, resulting in pathological circumstances like Alzheimer’s disease.	[88,89,90]
	Astrocytes can transport lactate to neurons, providing energy for brain activity and metabolism. Lactate decreases after peripheral surgery, which leads to a neuronal dysfunction.	[91,92,93]
	Astrocytes secrete glial cell line-derived neurotrophic factor (GDNF). Serum GDNF levels are lower in patients with PND than in healthy controls. Increased GDNF level reduce cognitive impairment following anesthesia and surgery.	[97,98]

## Data Availability

Not applicable.

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
