# Peer review of "The Role of Astrocytes in the Mechanism of Perioperative Neurocognitive Disorders"

_brainsci, 2022, doi:10.3390/brainsci12111435_

Round 1

Reviewer 1 Report

Comments and Suggestions for Authors

Authors put their significant contributions to compiling all the relevant references in the review. Most parts of the review written nicely and justified by references. Few of my suggestion mentioned below will be further helpful to the readers in the field. Also, in general, how PND further contributes in the development of long-term dementia or another neurodegenerative disease would be helpful.

 Introduction:

If the risk of getting PND is higher at an older age, is there any aging-related neurodegenerative report which further contributes to the PND? If so it would be nice to add them to this section.

In the processes of dementia during aging, cognitive impairments are usual, how authors can discriminate the cognitive impairments in those cases from PND?

Different phenotypes of reactive astrocytes and PND

Authors nicely described the role of A1 and A2 astrocytes, moreover, is this description related to older age PND or in general patients undergoing surgery? Whatever the case a clear description in this section would be helpful.

Astrocytes are involved in the development of PND partially by mediating neuroinflammation

Is surgery-induced pain/stress further contribute to the development of PND-induced impairment? If so kindly define it in this section.

Is there any time course study related to the development of PND-induced impairments and inflammation at the initial stage of surgery (when the pain was there) and by the time when surgery-induced pain and stress went in which direction PND-induced neuroinflammation or neurobehavioral impairment goes?

Abnormal astrocytes induce PND by regulating synapse formation, quantity, and  transmission of neurotransmitters

A brief, description of figure 2 in the figure legends should be helpful, also activation of astrocytes other than glucose (which contribute to this synaptic activation) need to be mention here to complete the figure.

Is there any report for the involvement of cholinergic and aminergic neurons in the PND development and phagocytic activity during long-term cognitive impairment? If so description of that in this section would be helpful.

Type of prostaglandin contributing to the calcium release needs to be mentioned in line 138.

What about neurogenesis, BDNF, and cytokines, they need to mention here in the figure.

Astrocytes maintain homeostasis of intracerebral microenvironment

Role of astrocytes end feet disruption, tight junction, and adherent junction proteins need to be describ in this section.

Authors limited the description only to aquaporin, connexin, and glymphatic system.  

Astrocytes have a lot of other roles, which contribute the BBB damage if any of them are related to PND development need to be mention here. Some references like 10.3389/fimmu.2017.00902, 10.1186/s12974-018-1374-3, 10.1213/ANE.0000000000004053 would be helpful for guidance on this.

In conclusion, authors need to think about rewriting this section.

Author Response

Authors put their significant contributions to compiling all the relevant references in the review. Most parts of the review written nicely and justified by references. Few of my suggestion mentioned below will be further helpful to the readers in the field. Also, in general, how PND further contributes in the development of long-term dementia or another neurodegenerative disease would be helpful.

 Introduction:

If the risk of getting PND is higher at an older age, is there any aging-related neurodegenerative report which further contributes to the PND? If so it would be nice to add them to this section.

R: We thank the reviewers for the kindly suggestion. We have added aging-related literature to confirm that older people are at greater risk of developing PND in the revised manuscript.

In the processes of dementia during aging, cognitive impairments are usual, how authors can discriminate the cognitive impairments in those cases from PND?

R : We thank the reviewers for the excellent suggestion. We have added the origin of the nomenclature of PND, which will facilitate the distinction between age-related cognitive impairment and PND.

Different phenotypes of reactive astrocytes and PND

Authors nicely described the role of A1 and A2 astrocytes, moreover, is this description related to older age PND or in general patients undergoing surgery? Whatever the case a clear description in this section would be helpful.

R:We thank the reviewers for the excellent suggestion. In the original manuscript we have listed the relationship between the A1 phenotype and PND and add the effect of A2 on PND in the revised manuscript.

Astrocytes are involved in the development of PND partially by mediating neuroinflammation

Is surgery-induced pain/stress further contribute to the development of PND-induced impairment? If so kindly define it in this section.

R:We thank the reviewers for the excellent suggestion. It is true that surgery-induced pain/stress has been reported in the literature to further promote PND, We have added relevant descriptions and literature in the revised manuscript.

Is there any time course study related to the development of PND-induced impairments and inflammation at the initial stage of surgery (when the pain was there) and by the time when surgery-induced pain and stress went in which direction PND-induced neuroinflammation or neurobehavioral impairment goes?

R:We thank the reviewers for the excellent suggestion. We have added relevant descriptions and literature in the revised manuscript.

Abnormal astrocytes induce PND by regulating synapse formation, quantity, and transmission of neurotransmitters

A brief, description of figure 2 in the figure legends should be helpful, also activation of astrocytes other than glucose (which contribute to this synaptic activation) need to be mention here to complete the figure.

R:We thank the reviewers for the excellent suggestion . we have made a small modification to Figure 2 in the revised manuscript.

Is there any report for the involvement of cholinergic and aminergic neurons in the PND development and phagocytic activity during long-term cognitive impairment? If so description of that in this section would be helpful.

R:We thank the reviewers for the excellent suggestion . We have added to the literature on the interaction of acetylcholine with astrocytes in the PND model in the revised manuscript. However, It is very unfortunate that we do not find any reports on the effect of aminergic neurons interacting with astrocytes on PND.

Type of prostaglandin contributing to the calcium release needs to be mentioned in line 138.

R:We thank the reviewers for the excellent suggestion . The type is prostaglandin E which has been shown in the article.

What about neurogenesis, BDNF, and cytokines, they need to mention here in the figure.

 R:We thank the reviewers for the excellent suggestion .We have added neurogenesis, BDNF ,IL-1,IL-6 in the revised Figure 1 and Figure 2.

Astrocytes maintain homeostasis of intracerebral microenvironment

Role of astrocytes end feet disruption, tight junction, and adherent junction proteins need to be describ in this section.

R:We thank the reviewers for the excellent suggestion .We have recognized that the description in the original version is incomplete, we have added relevant descriptions in the revised manuscript and have rewritten this section.

Authors limited the description only to aquaporin, connexin, and glymphatic system.  

Astrocytes have a lot of other roles, which contribute the BBB damage if any of them are related to PND development need to be mention here. Some references like 10.3389/fimmu.2017.00902, 10.1186/s12974-018-1374-3, 10.1213/ANE.0000000000004053 would be helpful for guidance on this.

In conclusion, authors need to think about rewriting this section.

R:We thank the reviewers for the excellent suggestion . We have recognized that the description in the original version is incomplete and have rewritten this section.

Reviewer 2 Report

Comments and Suggestions for Authors

This is a very good review of the available evidence of the role of astrocytes and PND. 

I have some comments:

Pg 2 ln93. Please extend the role of astrocytes on M1 polarization, including the importance. And relevance

Pg 3 ln 97, there are several promising pharmacological strategies. Either extend on these or elaborate on the one that is being mention, why it is being mentioned and why its is more important than other promising targets.

Kn 21, rephrase “common belief”. First, believes have no place in science and there are several studies that have shown what is stated.

Ln 153, the way this is framed, induces the reader to think AQP 4 channels are only present in astrocytes, which it is not the case. Please rephrase the paragraph.

I particularly enjoyed the summary table with the references listed. Congratulations.

Author Response

This is a very good review of the available evidence of the role of astrocytes and PND. 

I have some comments:

Pg 2 ln93. Please extend the role of astrocytes on M1 polarization, including the importance. And relevance

R:We thank the reviewers for the excellent suggestion .We have added M1 polarization -related descriptions to the revised manuscript.

Pg 3 ln 97, there are several promising pharmacological strategies. Either extend on these or elaborate on the one that is being mention, why it is being mentioned and why its is more important than other promising targets.

R:We thank the reviewers for the excellent suggestion .we have completely revised the section on neuroinflammation in the revised manuscript

Kn 21, rephrase “common belief”. First, believes have no place in science and there are several studies that have shown what is stated.

R:We thank the reviewers for the excellent suggestion and apologized for the

misunderstanding expression. “common belief” was revised to “There is no doubt about that”, pleased kindly find the revised manuscript.

Ln 153, the way this is framed, induces the reader to think AQP 4 channels are only present in astrocytes, which it is not the case. Please rephrase the paragraph.

 R:We thank the reviewers for the excellent suggestion . We have recognized that the description in the original version is inappropriate and have rewritten this section.

I particularly enjoyed the summary table with the references listed. Congratulations.

Reviewer 3 Report

Comments and Suggestions for Authors

Congratulations on a very interesting and timely review on the important role of astrocytes.  I do not have any substantive issues with the review as it stands.  Thanks again.  

Author Response

Congratulations on a very interesting and timely review on the important role of astrocytes.  I do not have any substantive issues with the review as it stands.  Thanks again.  

R: We thank the reviewer for the positive comments.
